# Health-Related Physical Fitness and Biochemical Parameters in Overweight Older People during Social Isolation Imposed by the COVID-19 Pandemic: A Longitudinal and Observational Study

**DOI:** 10.3390/ijerph21091161

**Published:** 2024-08-31

**Authors:** Marilene Ghiraldi de Souza Marques, Braulio Henrique Magnani Branco, Déborah Cristina de Souza Marques, Marielle Priscila de Paula Silva Lalucci, Victor Augusto Santos Perli, José Roberto Andrade do Nascimento, Pablo Valdés-Badilla, Daniel Vicentini de Oliveira

**Affiliations:** 1Interdisciplinary Laboratory of Intervention in Health Promotion, Cesumar Institute of Science, Technology and Innovation, Maringa 1610, PR, Brazil; marileneghiraldi@gmail.com (M.G.d.S.M.); marques.deborah@hotmail.com (D.C.d.S.M.); mariellepriscila@gmail.com (M.P.d.P.S.L.); 2Graduate Program in Health Promotion, Cesumar University, Maringa 1610, PR, Brazil; victoraperli@gmail.com (V.A.S.P.); daniel.vicentini@unicesumar.edu.br (D.V.d.O.); 3Physical Education College, Universidade Federal do Vale do São Francisco (UNIVASF), Petrolina 56304-917, PE, Brazil; jroberto.jrs01@gmail.com; 4Department of Physical Activity Sciences, Faculty of Educational Sciences, Universidad Católica del Maule, Talca 3530000, Chile; valdesbadilla@gmail.com; 5Sports Coaching Career, School of Education, Universidad Viña del Mar, Viña del Mar 2520000, Chile

**Keywords:** aging, sedentary behavior, glycemic control, health promotion

## Abstract

With COVID-19, evidence indicates that the elderly will have worse biochemical markers related to health in social isolation. The objective was to analyze the impacts on physical fitness and biochemical parameters of older adults’ health during COVID-19 social isolation. Quantitative, longitudinal, and observational study was conducted between 2020, 2021, and 2022. Thirty-three older adults of both sexes were evaluated. A sociodemographic questionnaire, biomarkers, and health-related physical fitness were used. Significant differences were observed for the sum of maximum isometric right and left handgrip strength, with a reduction in 2022 (*p* = 0.009); getting up and walking (*p* < 0.001), reduction in 2021 and 2022 (*p* < 0.05); elbow flexion and extension (*p* = 0.004), reduction in 2021 (*p* = 0.006); and sitting and standing (*p* = 0.002), reduction in 2022 (*p* = 0.003) and peak oxygen consumption (*p* < 0.001), reduction in 2021 and 2022 (*p* < 0.05). Differences were observed in fasting blood glucose (*p* < 0.001), with increase in 2021 and 2022 (*p* < 0.05), triglycerides (*p* < 0.001), with increase in 2021 and 2022 (*p* < 0.05), triglyceride–glucose index (*p* < 0.001), with increase in 2021 and 2022 (*p* < 0.05), triglyceride–glucose index with waist circumference (*p* < 0.001), with increase in 2021 (*p* < 0.001); and triglyceride–glucose index with body mass index (*p* < 0.001), with increase in 2021 (*p* < 0.001). However, no differences were observed between anthropometric and body composition (*p* > 0.05). Conclusions: Older people had changes in biochemical and physical fitness parameters related to health during the social isolation of the COVID-19 pandemic.

## 1. Introduction

In recent years, the infection caused by COVID-19 has caused one of the biggest public health problems in the world [1]. The virus characterized by respiratory tract infection can affect different areas (social, psychological, economic, and educational), causing cardiovascular and pulmonary sequelae, in addition to other related factors [2], becoming a significant mental health problem, being declared a pandemic by the world [3], leading to social isolation with changes in the social, health, and nutrition sphere of the population [1]. Lockdown during the COVID-19 pandemic not only altered the dynamics of social interactions but also had a profound impact on the mental health of the people who contracted or did not contract the SARS-CoV-2 virus in different symptoms [4].

Adhering to the imposed lifestyle change affected physical, psychological, and mental well-being [1]. This concern is mainly due to older people being more vulnerable to the consequences of COVID-19, both due to the aging process itself and because non-communicable diseases (NCDs) are risk factors for greater sequelae and severity of the infection [4], in addition to the COVID-19 pandemic result in recommendations for social distancing, which can lead to unhealthy behaviors, such as reduced physical activity levels and increased time spent using electronic devices, such as televisions, computers, and smartphones [1], causing inevitable and extreme changes in style and quality of life during the COVID-19 pandemic, negatively affecting the physical health of older people [5].

Indicators of the Triglyceride–Glucose Index (TyG), body mass index (BMI), waist circumference (WC), and visceral adiposity index (VAI) are essential for achieving glycemic control, being efficient in the control of diseases related to older age [6]. The aging process generates changes in physical fitness, which plays a fundamental role in the health status of older people, reflected in daily tasks, and influences the quality of life [7]. Therefore, other consequences and situations related to this period must be considered, such as the impact of social isolation during the COVID-19 pandemic.

Concerns following the COVID-19 pandemic have grown due to the rising health implications associated with increased sedentary behavior, which has led to a decline in the physical fitness of the population [1], directly related to the individual’s fragility [8]. Therefore, new studies analyzing the physiological and morphological changes during the COVID-19 pandemic in the health status of older people are necessary to manage the disease and provide strategies to deal with the population during this period. Continuous reassessment is essential to develop targeted health promotion interventions [9,10]. Understanding how age, sex, ethnicity, living conditions, social exposure, and chronic diseases influence long COVID allows us to adapt strategies and improve patients’ quality of life. In short, continuous assessment and monitoring of health indicators in post-COVID patients are essential to identify sequelae, guide treatments, and promote recovery means by guided physical exercises (aerobic, strength, or both) and specific nutritional plans focused on reduced possible inflammation and promoting muscle hypertrophy—if necessary [9,10,11].

Therefore, personalizing interventions based on these individual characteristics and using remote monitoring technologies can improve the monitoring and effectiveness of applied therapies [4,11] and could improve health status. Investigating the health consequences of older people at different times is essential to prevent health-related consequences and promote active and healthy aging [8]. Therefore, analyzing which variables require attention in the post-pandemic period is essential to promoting people’s health directly or indirectly impacted. Therefore, this study aimed to analyze the impacts of health-related physical fitness and biochemical parameters in overweight older people during social isolation imposed by the COVID-19 pandemic. A hypothesis is a probability that older people will have worse health-related parameters of health and biochemical markers due to social isolation.

## 2. Materials and Methods

### 2.1. Study Design

This quantitative, longitudinal, and observational study was carried out over three years (2020, 2021, and 2022) and assessed health-related physical fitness and biochemical parameters in overweight older people.

### 2.2. Participants

In order to participate in the study, 40 older people of both sexes were invited to participate in an outreach project at a university northwest of Paraná, Brazil, between 2020 and 2022. Participants were recruited in Basic Health Units in the same region and clinics around the University Center by publicizing the project with posters, pamphlets, and advertising on websites, television, and radio.

### 2.3. Inclusion and Exclusion Criteria

Inclusion criteria were considered: being 60 years old or over; having participated in the university’s health promotion laboratory, with a project for older people in 2019; having preserved cognitive, speech, and hearing capacity; and having physical conditions to take physical fitness assessments. Exclusion criteria: older people who did not participate in the assessments carried out in the period in question (2020, 2021, and 2022); individuals with physical limitations to perform the requested evaluations; and those who use drugs to regulate appetite or psychotropic. Additionally, the present study followed the statements of Strengthening the Reporting of Observational Studies in Epidemiology—STROBE [12]. The Local Research Ethics Committee approved the study under number 3.373.307/2019. The project wholly followed resolution 466/12 of the Ministry of Health. All participants were invited to sign the Informed Consent Form (ICF). Therefore, in the last data collection, 2022, thirty-three older people were evaluated, according to the flowchart (Figure 1).

Older people were recruited at the beginning of 2020 and assessed at three different times: 2020 (before the pandemic), 2021 (during the pandemic), and 2022 (post-social isolation), in the Health Promotion Laboratory of the Educational Higher Institution. After recruiting participants, an initial meeting was scheduled to explain the technical procedures of the research/extension. Subsequently, individual assessments were scheduled with the older people on pre-determined days. The study followed the order: (i) explanations about the objective of the study; (ii) filling out the Informed Consent Form (ICF); (iii) interview carried out by a health professional (completing an instrument to verify the socioeconomic, health, and nutritional conditions of the participants); (iv) collection of blood tests; (v) anthropometry and body composition; (vi) physical fitness assessments. The same researchers repeated the process over the three years of the study.

### 2.4. Interview—Healthcare Professional

In order to evaluate the sociodemographic and general health profile, a questionnaire prepared by the authors was used, with questions regarding age group, sex, race/ethnicity, marital status, occupational status, education, existing NCDs, history clinical status, medication use, self-perception of health and whether they contracted COVID-19.

### 2.5. Blood Collections

Blood analyses were performed on a pre-determined day, in the morning, after an overnight fast of approximately 12 h. The study collected samples from older people between 2020, 2021, and 2022. All analyses were carried out by biomedical professionals duly registered with the Regional Council of the Profession. The participants’ fasting blood glucose and triglycerides (TG) were analyzed. After asepsis of the arm, a puncture was performed in the antecubital veins of those assessed. Biomarkers of fasting blood glucose and triglycerides (TG) were analyzed. After collection, the samples were centrifuged at 3.600 rpm for 10 min at room temperature (24 °C) to separate serum and plasma. Analyses were performed in triplicate using Siemens^®^ reagents (Frimley, Camberley, UK) according to the specifications established by the manufacturer. The tubes used were vacuum tubes (Becton Dickinson—Vacutainer^®^, Plymouth, UK) for all collections, being: the tube with potassium fluoride for the analysis of fasting blood glucose (fluoridated plasma), and the tube with clot activator (silica) for the analysis of TG (serum). Subsequently, Siemens equipment was used for biochemical analyses (Advia 1800 Chemistry Analyzer^®^, Siemens Healthcare Diagnostics, Deerfield, IL, USA) and classified according to the cutoff points of the Brazilian Society of Cardiology [13] and Brazilian Society of Diabetes [14]. In addition, the Brazilian Diabetes Society (SBD) adopted fasting glucose cutoff points < 100 mg/dL for normality; pre-diabetes or increased risk 100 mg/dL and < 126 mg/dL; and established diabetes 126 mg/dL. Triglycerides were considered below 150 mg/dL for normality and above 150 mg/dL for cardiovascular risk.

### 2.6. Glycemic Control Indicators

Based on the results found in blood tests, it was possible to calculate glycemic control indicators using triglyceride values [6] by the formula:

Triglyceride–Glucose Index (TyG Index) = (fasting triglycerides [mg/dL] × Fasting blood glucose [mg/dL]/2)

With the result found, anthropometric measurements were combined with the TyG index to assess cardiometabolic risk, which allowed the calculation of the following composite indexes:TyG-WC = TyG Index × WC (waist circumference)
TYG-BMI = TyG Index × BMI (body mass index)

### 2.7. Anthropometry and Body Composition

Height was measured using a Sanny^®^, São Bernardo do Campo, SP, Brazil, stadiometer, following the standardization proposed by Lohman, Roche, and Martorell [15]. Body mass (kg) was measured using the InBody 570^®^ equipment (InBody^®^, Body Composition Analyzer, Seoul, Republic of Korea).

With height and body mass data, BMI was calculated by dividing body weight by height squared (BMI = W/(H^2^). With the results found for BMI and age of each participant, the nutritional status of the older people was assessed, and they were classified as within normal limits, overweight, and obese, according to WHO criteria [16]. InBody 570^®^ bioimpedance was used to assess body composition with which the following variables were collected: (i) Body mass (kg), (ii) Index of body mass (BMI, kg/m^2^), (iii) fat-free mass (FFM, in kg); (iv) lean mass (LM, in kg); (v) skeletal muscle mass (SMM, in kg); (vi) fat mass (FM, in kg) and (vii) body fat percentage (BFP, in %). Additionally, for the body composition assessments, the participants were asked to follow the following procedures: (i) fasting from solids and liquids for approximately 12 h; (ii) not using diuretic substances in the 24 h before the procedure; (iii) do not do moderate- or high-intensity physical exercise on the day before the assessments; (iv) avoid the consumption of caffeine-based drinks for the previous 12 h; (v) urinate 30 min before taking the exam; (vi) do not use any earrings, metal, or accessory at the time of collection; and, finally, (vii) participants were asked to wear light clothing at the time of the assessment [17].

Additionally, waist circumference (WC, in cm), hip (HC, in cm), neck (NC, in cm), and arm (AC, in com using a flexible measuring tape (Cescorf^®^, Porto Alegre, Brazil) were measured, with a measuring capacity of 2 m and accuracy of 0.1 cm. With the values found, the waist-to-hip ratio (WHR) was measured by the division of WC by HC [14].

### 2.8. Physical Fitness Assessments

The sit-and-reach test assessed the flexibility of the back of the trunk and lower limbs using a Wells bench (model: BW2002, Sanny^®^) and a mat. For this test, individuals were instructed to sit barefoot on a mat with their lower limbs extended and the soles of their feet placed against the front surface of the device. From this position, participants should flex their torso by sliding their fingers along the ruler to the maximum point of their reach on the bench, with a scaled marking in centimeters used to assess this parameter. Each participant performed three attempts, with the highest value expressed in centimeters being considered [18].

Participants also underwent the Senior Fitness Test (SFT) battery of tests to measure physical fitness. The SFT was developed and validated for older people by researchers Jessie Jones and Roberta Rikli [19] from California State University, California, United States, as part of the Life Span Assessments Project or Fullerton Functional Fitness Test. This method evaluates the main parameters of older people’s functional capacity and dependence [16]. The evaluations followed the order: (i) Arm Curl Test; (ii) Sit–Stand test; (iii) Sitting, walk 2.44 m, and sit back down; (iv) 6-min walk test.

The elbow arm curl test was used to assess muscle strength and endurance of the upper limbs. The number of elbow pushups gives this test a weight of 2 kg for females and 4 kg for males for 30 s [19]. The participant was positioned in a chair with the body close to the edge, aligned with the back straight and feet on the floor. With the dominant hand, he was instructed to hold the dumbbell using a handshake grip; at the indicative signal, the participant rotated his palm upwards while flexing the arm in the full range of motion and returning the arm to an utterly extended position.

In addition to the elbow arm curl test, the maximal isometric handgrip strength test was applied using the Takei dynamometer model TKK 5101 (Takei Scientific Instruments, Tokyo, Japan) with a capacity of 0 to 100 kg per force (kg/f). In this test, participants were instructed to sit comfortably, with shoulders slightly adducted, the elbow flexed at an angle of 90°, and the forearm in a neutral position, with wrist positioning varying between an angle of 0 to 30°. The test was performed three times on each hand, lasting 3 s and an interval of 1 minute between each attempt, with the highest value obtained being considered. The cutoff point proposed by Bohannon et al. [20] was considered for the classification.

The movement chair stand test is a prerequisite for mobility and functional independence [21]. The assessed began the test sitting in a chair with his back supported, feet on the floor, and arms crossed over the chest. At the signal, the participant was instructed to stand and then return to the sitting position, in which he performed as many of these movements as possible for 30 seconds. At the end, the number of repetitions was noted on the evaluation form. For this test, a chair with a backrest and without arms was used, with a height of approximately 43 cm and a timer [21].

The Sitting, walk 2.44 m, and sit back down test was also used to assess the lower limbs’ physical mobility (speed, agility, and dynamic balance). The test consists of getting up from a chair (approximately 43 cm), walking to a straight line 2.44 m away (at a self-selected but safe pace), turning around, walking back, and sitting down again. The shorter the time used, the better the performance in the test consists of analyzing the time (seconds) it takes the participant to walk 2.44 m from sitting until returning to the same position.

The 6-minute walk test (6MWT) aimed to evaluate the relationship between physical fitness and maximum oxygen consumption. The test followed the guidelines of the American Association of Cardiovascular and Pulmonary Rehabilitation, in which the older people walked on a circular track measuring 45.60 m for 6 min, counting the highest number of laps completed in 6 min, without verbal encouragement from the researchers [22]. Before exercising, those evaluated were instructed to rest for 10 min to measure blood pressure and wear comfortable clothes and shoes to perform the test. After the exercise, blood pressure was also measured before and after the test. The calculation of peak oxygen consumption was carried out using the equation by Cahalin et al. [22]:VO_2_peak (mL·kg^−1^·min^−1^) = (0.02 × distance [m]) − (0.191 × age [years]) − (0.07 × body mass [kg] + (0.09 × height [cm]) + (0.26 × RPP [10 − 3]) + 2.45.
where the abbreviations of the VO_2_peak predictive equation are: (a) m = distance in meters; (b) age in years; (c) body mass in kilograms (kg), and (d) RRP = double product that was calculated by heart rate multiplied by systolic blood pressure (SBP) (millimeter of mercury, mmHg). During the entire time, individuals were monitored using an oximeter to assess oxygen saturation and a heart monitor (Polar watch model FT1, Kempele, Finland) to check heart rate and, at the end of the test, Blood Pressure (BP). To familiarize themselves with the tests, participants had a moment of explanation and experimentation.

### 2.9. Data Analysis

Data were analyzed using SPSS software, version 25.0, using descriptive and inferential statistics, with frequency and percentage as descriptive measures for categorical variables; for numerical variables, the normality of the data was initially verified using the Shapiro–Wilk test, and the mean and standard deviation were used as measures of central tendency and dispersion. Bootstrapping procedures (1000 re-samples; 95% CI BCa) were also carried out to obtain significant results reliability, correct possible deviations from the sample distribution’s normality, and present a 95% confidence interval for the means. A one-way repeated-measures Analysis of Variance (ANOVA) was conducted to compare physical fitness tests and biochemical parameters over 2020, 2021, and 2022, followed by Bonferroni post hoc tests. Effect sizes (ƞ^2^p) were classified according to Richardson [23] as follows: 0.0099 (small), 0.0588 (moderate), and 0.1379 (large). A significance level of *p* < 0.05 was used for all analyses.

## 3. Results

Twenty-three older females and 10 males participated in the study. According to the results in Table 1, it was found that the majority of participants were in the age group of 65 to 70 years (61.8%), had a monthly income of one to three minimum wages (52.9%), were white (76.5%), retired (91.2%), reported having a chronic NCD (91.2%), used medication (94.1%), and had a negative COVID-19 test (70.6%). It was also observed that most older people had at least completed high school (73.5%).

The data presented in Table 2 refer to the collection of physical fitness assessments carried out between 2020, 2021, and 2022.

Table 2 presents the responses of the physical and cardiorespiratory fitness parameters during the period. Significant differences were observed for the sum of the maximum isometric handgrip strength of the right and left sides (*p* < 0.001; ŋ^2^*p* = 0.19—large), with a significant reduction in 2022 (*p* = 0.009); TUG (*p* < 0.001; ŋ^2^*p* = 0.42—large), with a reduction in 2021 (*p* = 0.004) and 2022 (*p* < 0.001); elbow flexion and extension (*p* = 0.004; ŋ^2^*p* = 0.16—moderate), with a significant reduction in 2021 (*p* = 0.006) and sit-stand test (*p* = 0.002; ŋ^2^*p* = 0.17—large), with a significant reduction in 2022 (*p* = 0.003); and VO_2_peak (*p* < 0.001; ŋ^2^*p* = 0.38—large), with a reduction in 2021 (*p* < 0.001) and 2022 (*p* < 0.001). No significant differences were observed for flexibility, maximal isometric handgrip strength for right and left sides, HRmax, final SBP, and final DBP (*p* > 0.05). Table 3 presents the descriptive analyses of anthropometric parameters and body composition of the older people participating in the study between 2020, 2021, and 2022.

The results in Table 3 indicate no significant difference in anthropometric and body composition parameters over the three years of monitoring (*p* > 0.05). Table 4 presents the biochemical parameters between 2020 and 2022 of older people participating in the study.

Table 4 presents the biochemical parameters. Significant differences were observed for fasting blood glucose (*p* < 0.001; ŋ^2^*p* = 0.53—large), with an increase in 2021 (*p* < 0.001) and 2022 (*p* = 0.019); triglycerides (*p* < 0.001; ŋ^2^*p* = 0.58—large), with an increase in 2021 (*p* < 0.001) and 2022 (*p* = 0.002); TyG (*p* < 0.001; ŋ^2^*p* = 0.65—large), with an increase in 2021 (*p* < 0.001) and 2022 (*p* < 0.001); TyG.WC (*p* < 0.001; ŋ^2^*p* = 0.53—large), with an increase in 2021 (*p* < 0.001); and TyG.BMI (*p* < 0.001; ŋ^2^*p* = 0.50—large), with an increase in 2021 (*p* < 0.001).

## 4. Discussion

The study aimed to analyze the impacts of health-related physical fitness and biochemical parameters in overweight older people during social isolation imposed by the COVID-19 pandemic. The central outcomes revealed several significant changes in different health parameters, highlighting the influence of social isolation on older people.

### 4.1. Changes in Muscle Strength

One of the most relevant findings was the reduction in maximum isometric handgrip strength, both in the right and left hands, in 2022. This decline can be attributed to the restrictions and lifestyle changes imposed by the pandemic, which resulted in decreased physical activity and increased social isolation. Previous studies support this observation, indicating that the interruption of exercise routines and the psychological impact of the pandemic contributed significantly to the loss of muscle strength in older people [23]. The decrease in muscle strength in the upper and lower limbs in 2022 reinforces the continued negative impacts of the pandemic on musculoskeletal health.

The lack of access to exercise spaces and less participation in daily activities probably compromised the maintenance and development of muscle strength in both body regions [24]. Coronavirus infection may also have contributed to sarcopenia in older people by exacerbating muscle loss [25]. Studies such as that by Bravalhieri et al. [26] showed a significant reduction of 30.2% in muscle strength after five months of social isolation, while Cezón-Serrano et al. [27] recorded a decline in muscle strength, especially in the biceps brachii, after 15 weeks of isolation. These findings highlight the need for rehabilitation and health promotion strategies to preserve muscle strength during and after isolation.

### 4.2. Fall Risk

The increased risk of falling, assessed by the Sitting, walking 2.44 m, and sit back down test 2022, may be linked to the interruption of regular physical activities and social distancing. The reduction in mobility and balance, resulting from the reduction in exercise and social interaction, may have contributed to this increase [28]. Anxiety associated with the health crisis may also have exacerbated this problem, resulting in a longer time to perform the Sitting, walking 2.44 m, and sit back down test [24]. Hoffman et al. corroborated the relationship between sedentary behavior, social isolation, and increased fall risk [29], who found a higher fall risk in individuals with compromised mobility and who were socially isolated. These results highlight the importance of developing specific strategies to prevent falls and addressing physical, emotional, and social health [30].

### 4.3. Cardiorespiratory Fitness

The reduction in VO_2_peak in 2021 indicates an adverse impact of the pandemic on the aerobic capacity of older people. This decline reflects the body’s maximum capacity to use oxygen during exercise and suggests a harmful interference with regular physical activity practice and cardiovascular conditions [31]. Silva et al. indicate that some months of lockdown could substantially reduce VO_2_peak, even with home tele-exercises during COVID-19 [30]. Thus, the reduced VO_2_peak observed in 2021 refers to low stimulus during the day; these results could be expected since the movement of people has decreased significantly, and even the stimulus cited by da Silva et al. was not enough to promote maintaining this capacity. Telerehabilitation and exercise strategies could be tested to develop new possibilities for maintaining health for the population, especially older people.

Over the past few years, COVID-19 has had a lasting impact on public health, necessitating the development of effective strategies for controlling and treating affected patients [9,32]. Persistent symptoms such as fatigue, respiratory difficulties, and a reduced ability to carry out daily activities are frequently observed in individuals who have recovered from the acute phase of the infection [10,11]. Research, such as that conducted by Huang et al. [33], has revealed that a significant number of patients continue to experience debilitating symptoms. During this period, rehabilitation programs focusing on cardiorespiratory and neuromuscular recovery emerged, aiming to help patients return to their daily activities and improve their quality of life [9]. Physical exercise has been increasingly recognized as an effective strategy to mitigate the adverse effects of COVID-19, as highlighted by Jimeno-Almazán et al. [34].

Ongoing monitoring of patients has proven to be essential, as physical fitness is crucial for performing daily activities and maintaining good work conditions. Additionally, the uncertainty surrounding the long-term effects of COVID-19 on various organ systems has underscored the need for constant vigilance and an interdisciplinary approach to patient care [11]. In summary, the evolution of COVID-19 treatment strategies from 2020 to 2022 reflects a progressive shift towards a more integrated and effective management of the disease’s long-term effects, focusing not only on restoring patients’ physical capacities but also on prevention, as highlighted by Perli et al. [11], ensuring that patients can resume their lives with health and vitality.

### 4.4. Stability of Some Measured Parameters

Although no significant changes were observed in flexibility, HRpeak, SBP, and DBP, as well as in anthropometric parameters and body composition over the three years of monitoring, these findings suggest stability in various physical and physiological aspects during the period studied; this can be attributed to health promotion initiatives implemented before the pandemic [24]. Additionally, the absence of anthropometric and body composition differences could also be related to food reeducation in older people, considering the responses of Marques et al. [35] with the same population that participated in the present study. However, making inferences about this aspect is impossible since this instrument was not applied in the present study.

### 4.5. Metabolic Changes

The increased fasting blood glucose and triglycerides in 2021 and 2022 indicate possible metabolic changes. Changes in eating patterns, increased stress, and reduced physical activity may have contributed to these increases [33]. Social isolation can lead to binge eating, anxiety, and stress, which interfere with homeostatic balance and result in health problems [34]. The increase in TyG.WC and TyG.BMI indexes in 2021 suggest changes in glycemic homeostasis and the insulin-resistant profile. These indexes, which combine fasting glucose with WC and BMI, are indicators of insulin resistance, a risk factor for type 2 diabetes and cardiovascular disease [6]. Changes in eating patterns and lack of physical activity during the pandemic likely contributed to these increases [33].

### 4.6. Implications and Future Needs

This study underscores the critical importance of intervention strategies to enhance muscle strength and cardiovascular and metabolic health among older adults, particularly during health crises. The COVID-19 pandemic has illuminated the vulnerability of older individuals to abrupt lifestyle changes, emphasizing the necessity for developing intervention plans that can be swiftly implemented in similar future scenarios.

### 4.7. Study Limitations

Despite the significant findings, this study has several limitations. One major limitation is the lack of control over participants’ daily physical activity. While the study offers valuable insights into the impact of lockdown on physical health, a more comprehensive understanding of the changes would require examining how isolation influenced the type, quantity, and intensity of physical activities performed. The absence of post-social isolation monitoring further hinders the assessment of long-term effects and the resumption of physical activities in both indoor and outdoor environments. Additionally, the lack of dietary control and daily physical activity monitoring limits our understanding of changes in eating habits that may have influenced the results. Another critical limitation is the absence of a control group, which could have significantly strengthened the study by providing a baseline for comparison and enhancing the accuracy of the results. The omission of a control group thus represents a notable limitation of the current study.

## 5. Conclusions

The older people suffered changes in biochemical and physical fitness parameters related to health during the social isolation imposed by the COVID-19 pandemic. Therefore, reversing these factors becomes extremely important. Encouraging interventionist activities with older people is essential to promote improvements in quality of life, with programs aimed at public health and the like. Therefore, it is recommended that a multidisciplinary approach be applied with regular physical activity, healthy eating, and psychoeducation in these older people and that the results be reversed in the three different moments.

## Figures and Tables

**Figure 1 ijerph-21-01161-f001:**
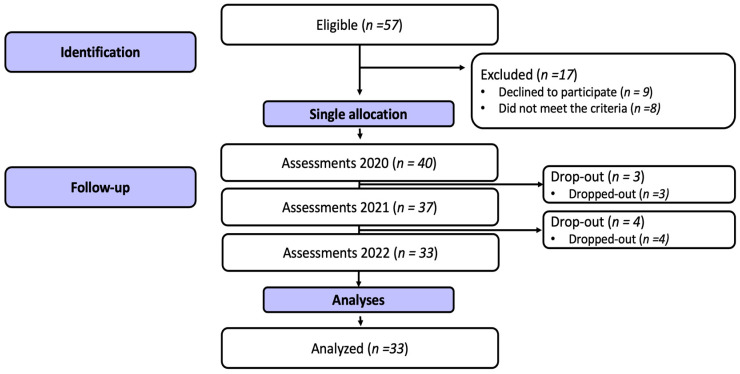
Flowchart of the present study.

**Table 1 ijerph-21-01161-t001:** Sociodemographic profile of participants in the last assessments (2022).

Variables	*ƒ*	%
Sex		
Female	23	69.70
Male	10	30.3
Age group		
60 to 65 years	6	17.6
65 to 70 years	21	61.8
70 to 75 years	4	14.7
75 to 80 years	2	5.9
Monthly Income		
Up to 1 MW	0	0.0
1 to 3 MW	6	18.18
3 to 6 MW	18	54.54
6 to 9 MW	6	18.18
9 a 12 MW	3	9.10
Education		
Elementary education I	8	24.24
Elementary education II	0	0.0
Incomplete high school	0	0.0
Complete high school	11	33.2
Incomplete university education	3	9.10
Complete university education	11	30.2
Color		
White	25	75.7
Black	8	21.2
Pathology		
Yes	32	96.97
No	1	3.03
Retirement		
Yes	30	90.90
No	3	9.10
Medications		
Yes	31	93.4
No	2	6.6
Covid testing		
Yes	10	30.3
No	23	69.7

Note: MW: minimum wage (s).

**Table 2 ijerph-21-01161-t002:** Physical fitness assessments were performed between 2020, 2021, and 2022 on the older people participating in this study.

Physical Fitness Tests/Measurements	2020	2021	2022
Maximum isometric handgrip strength— right (kgf)	28.7 ± 9.6 (25.4–32.1)	28.9 ± 9.6 (25.6–32.2)	27.4 ± 9.0 (24.3–30.5)
Maximal isometric handgrip strength— left (kgf)	27.1 ± 9.0 (24.0–30.3)	28.1 ± 9.0 24.9–31.2)	25.7 ± 8.2 (22.8–28.5)
Sum of the maximum isometric handgrip strength of right and left sides (kgf) ^‡^	55.9 ± 18.4 (49.5–62.3)	57.0 ± 18.5 (50.5–63.4)	53.1 ± 17.0 (47.2–59.0)
Flexibility (Wells bench, cm)	24.3 ± 7.7 (21.6–27.0)	23.9 ± 7.6 (21.2–26.5)	23.1 ± 7.8 (20.4–25.8)
Sitting, walking 2.44 m, and sitting back down test ^†^	409.9 ± 60.8 (388.3–431.4)	374.7 ± 59.5 (353.9–395.4)	444.4 ± 66.7 (420.8–468.0)
Arm Curl Test (reps/min) *	19.5 ± 4.4 (17.9–21.0)	17.0 ± 4.6 (15.4–18.7)	17.6 ± 4.4 (16.1–19.1)
Sit–Stand test (reps/min) ^‡^	18.3 ± 5.4 (16.4–20,2)	16.7 ± 4.4 (15.2–18.2)	15.6 ± 3.7 (14.3–16.9)
6MWT			
VO_2_ peak (mL/kg/min) *	14.8 ± 2.5 (13.9–15.7)	13.7 ± 2.7 (12.7–14.6)	13.5 ± 1.8 (12.8–14.1)
Distance (m)	578.5 ± 91.7 (546.5–610.5)	536.2 ± 94.4 (503.3–569.1)	513.8 ± 63.1 (491.4–536.2)
HRpeak (bpm)	119 ± 21 (112–127)	111 ± 22 (103–118)	115 ± 17 (109–121)
Final SBP (mmHg)	160.7 ± 22.8 (152.8–168.7)	154.5 ± 17.9 148.3–160.8)	155.4 ± 18.1 (149.1–161.8)
Final DBP (mmHg)	77.7 ± 12.8 (73.2–82.1)	79.4 ± 9.8 (76.0–82.8)	75.3 ± 10.5 (71.6–79.0)

Note: Data described by the mean, standard deviation (±), and confidence interval at 95%; HRpeak = heart rate peak; DBP = diastolic blood pressure; SBP = systolic blood pressure; 6MWT = 6-min walk test; VO_2_ = oxygen consumption; * = significant reduction in 2021 (*p* < 0.05); ^‡^ = significant reduction in 2022 (*p* < 0.05); ^†^ = significant difference in 2021 and 2022 (*p* < 0.05).

**Table 3 ijerph-21-01161-t003:** Anthropometric and body composition parameters of older people participating in this study between 2020, 2021, and 2022.

Anthropometric and Body Composition Parameters	2020	2021	2022
Body mass (kg)	79.3 ± 21.0 (72.0–86.6)	78.2 ± 19.7 (71.2–85.2)	79.2 ± 20.1 (72.2–86.2)
BMI (kg/m^2^)	30.5 ± 6.1 (28.3–32.6)	30.9 ± 6.3 (28.7–33.1)	30.7 ± 6.2 (28.6–32.9)
FFM (kg)	46.6 ± 9.6 (43.2–50.0)	45.9 ± 9.5 (42.5–49.2)	46.1 ± 9.1 (42.8–49.3)
LM (kg)	43.9 ± 9.1 (40.7–47.1)	43.2 ± 8.9 (40.1–46.4)	43.4 ± 8.5 (40.3–46.4)
SMM (kg)	25.5 ± 5.8 (23.4–27.5)	25.0 ± 5.5 (23.0–27.0)	25.0 ± 5.4 (23.1–27.0)
FM (kg)	31.5 ± 12.1 (27.3–35.7)	32.5 ± 12.3 (28.2–36.8)	32.1 ± 12.2 (27.9–36.4)
BFP (%)	38.9 ± 7.4 (36.3–41.4)	39.8 ± 7.6 (37.2–42.4)	39.0 ± 7.7 (36.3–41.7)
WC (cm)	102.5 ± 14.0 (97.6–107.3)	102.9 ± 14.8 (97.7–108.1)	102.9 ± 13.8 (98.0–107.7)
HC (cm)	107.1 ± 13.3 (102.4–111.7)	105.7 ± 14.0 (100.9–110.6)	106.0 ± 12.5 (101.6–110.3)
WHR	0.94 ± 0.1 (0.91–0.97)	0.93 ± 0.1 (0.90–0.96)	0.93 ± 0.1 (0.89–0.96)
NC (cm)	37.5 ± 3.9 (36.1–38.9)	37.5 ± 4.5 (35.9–39.0)	37.7 ± 4.3 (36.2–39.2)
AC (cm)	33.9 ± 5.0 (32.2–35.6)	34.0 ± 4.4 (32.4–35.5)	34.2 ± 4.2 (32.8–35.7)
CCALF (cm)	37.2 ± 4.1 (35.8–38.7)	37.9 ± 4.7 (36.2–39.6)	37.3 ± 4.5 (35.8–38.9)

Note: Data described by the mean, standard deviation (±), and confidence interval at 95%; WC = waist circumference; HC = hip circumference; WHR = waist-to-hip ratio; WC = neck circumference; AC = arm circumference; CCALF = calf circumference; BMI = body mass index; FM= fat mass; FFM = fat-free mass; LM = lean mass; SMM = musculoskeletal mass; BFP = body fat percentage.

**Table 4 ijerph-21-01161-t004:** Biochemical parameters of older people participating in this study between 2020, 2021, and 2022.

Biochemical Parameters	2020	2021	2022
Fasting glucose (mg/dL) ^†^	102.2 ± 17.3 (96.0–108.5)	116.1 ± 21.7 (108.1–124.0)	127.0 ± 20.2 (119.8–134.3)
Triglycerides (mg /dL) ^†^	96.0 ± 39.7 (81.9–110.1)	130.1 ± 48.3 (113.7–147.3)	154.9 ± 58.4 (134.5–175.3)
TyG †	5360.5 ± 3006.0 (4294.6–6426.3)	7547.8 ± 3411.7 (6296.4–8799.3)	10,204.1 ± 4133.2 (8761.9–11,646.2)
TyG.WC *	558,333.0 ± 316,706.7 (456,475.1–768,195.3)	887,547.6 ± 524,402.5 (725,281.6–1,206,773.9)	1,057,255.6 ± 461,500.2 (896,239.7–1,218,280.8)
TyG.BMI *	167,009.9 ± 97,417.6 (132,467.1–201,552.7)	268,877.2 ± 162,765.5 (211,163.1–326,591.3)	319,394.0 ± 161,171.3 (263,158.7–375,629.3)

Note: Data described by the mean, standard deviation (±), and confidence interval at 95%; WC = waist circumference; BMI = body mass index; TyG = triglyceride–glucose index. * = significant increase in 2021 (*p <* 0.05); ^†^ = significant increase in 2021 and 2022 (*p <* 0.05).

## Data Availability

The data generated during the study will be informed when requested.

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
