# Peer review of "Health-Related Physical Fitness and Biochemical Parameters in Overweight Older People during Social Isolation Imposed by the COVID-19 Pandemic: A Longitudinal and Observational Study"

_ijerph, 2024, doi:10.3390/ijerph21091161_

Round 1
Reviewer 1 Report
Comments and Suggestions for Authors
Dear authors,
your work about " Health-related physical fitness and biochemical parameters in overweight older people during social isolation imposed by the COVID-19 pandemic: a longitudinal and observational study" with the aim of to analyze the impacts of health-related physical fitness and biochemical parameters in overweight older people during social isolation imposed by the COVID-19 pandemic.” has some interesting points to the literature. My suggestions are:
a) Introduction must present some background of the need of this study and some sentences about the effects of exercise in this period of covid.
b) In methods, its not clear the inclusion and exclusion criteria used. If the pearson had some injury would be accepted to your research? Please describe with more specificity all your criteria.
c) During covid how did you collect data? There were a lot of restrictions how did you acess those people? How can you guarantee that all participants performed the same daily routine (physical activity, nutrition….)
d) Add IC and p values of all variables of Table 2.
e) Your discussion needs substancial improvemments. You do not discuss your data with several similar studies of this topic (there are a lot….) This is your weak point of the manuscript submitted. One example “The reduction in VO2peak in 2021 indicates an adverse impact of the pandemic on the aerobic capacity of older people. This decline reflects the body's maximum capacity to use oxygen during exercise and suggests a harmful interference with regular physical activity practice and cardiovascular conditions [26]. The literature indicates that even one year of social isolation in young adults can substantially reduce VO2peak [27].” Where is the discussion of all three- years? Where is the discussion with other studies? You used a study of young adults….
f) Where is the control group?
Comments on the Quality of English LanguageNeed improvemments in all sections
Author Response
COVER LETTER
Dear Editor,
We would like to thank you for the opportunity to respond to the reviewers. We appreciate their efforts and consideration of all comments, and we believe that their feedback strengthened our manuscript. Our responses to their suggestions are below (answered point by point) and were highlighted in yellow in the original article.
Sincerely,
The authors.
Since social isolation is not specific to a pandemic, the results of this study may be more relevant. However, the authors failed to make this reference.
R: Lockdown during the COVID-19 pandemic not only altered the dynamics of social interactions but also had a profound impact on the mental health of the people who contracted or did not contract the SARS-CoV-2 in different symptoms [4] – lines 45, 46, and 47.
4. Ryal, JJ.; Perli, V.A.S.; Marques, D.C.d.S.; Sordi, A.F.; Marques, M.G.d.S.; Camilo, M.L.; Milani, R.G.; Mota, J.; Valdés-Badilla, P.; Magnani Branco, B.H. Effects of a Multi-Professional Intervention on Mental Health of Middle-Aged Overweight Survivors of COVID-19: A Clinical Trial. Int. J. Environ. Res. Public Health 2023, 20, 4132. doi.org/10.3390/ijerph20054132
The study seems very interesting, but it should have addressed and discussed the lack of control over daily physical activity to understand if there was a change in the type of physical activities, their quantity, and possibly their intensity. Some declines in physical performance may be associated with health changes related to the aging process. Some of the changes are significant but of little magnitude. It is understood that not everything can be evaluated, but the authors should mention these issues.
R: Thank you for your recommendation. We have added a paragraph with this information in the study's limitations.
Despite the significant findings, this study has several limitations. One major limitation is the lack of control over participants' daily physical activity. While the study offers valuable insights into the impact of lockdown on physical health, a more comprehensive understanding of the changes would require examining how isolation influenced the type, quantity, and intensity of physical activities performed. The absence of post-social isolation monitoring further hinders the assessment of long-term effects and the resumption of physical activities in both indoor and outdoor environments. Additionally, the lack of dietary control and daily physical activity monitoring limits our understanding of changes in eating habits that may have influenced the results - lines 412-421.
Despite the well-designed study and the interesting sample, the authors mentioned several other studies in the introduction with similar purposes. I am concerned about the lack of novelty in the study, apart from the Brazilian sample. The authors should emphasize what new insights this study provides.
R: Thank you for your recommendation. We added some sentences to add this information: This study underscores the critical importance of intervention strategies to enhance muscle strength, cardiovascular health, and metabolic health among older adults, particularly during health crises. The COVID-19 pandemic has illuminated the vulnerability of older individuals to abrupt lifestyle changes, emphasizing the necessity for developing intervention plans that can be swiftly implemented in similar future scenarios – Lines: 407 – 411.
AND
Telerehabilitation and exercise strategies could be tested to develop new possibilities for maintaining health for the population, especially older people – Lines: 360 and 361.
The manuscript should be revised by a native English-speaking scientist to ensure all the information in the study is clearly understood. For example, in the 3rd sentence of the abstract, "Quantitative, longitudinal, and observational study between 2020, 2021, and 2022," it seems that "conducted" was omitted.
R: Thank you for your feedback. We sent this article again to ensure the accuracy of the English.
4rd paragraph 1st sentence: "Concerns after the COVID-19 pandemic due to the emerging numbers of health implications increased sedentary behavior and, consequently, decreased the population's physical fitness". The sentence is confusing and should be rewritten.
R: To enhance the clarity of the sentence, it has been revised as follows:
"Concerns following the COVID-19 pandemic have grown due to the rising health implications associated with increased sedentary behavior, which has led to a decline in the physical fitness of the population…"
Introduction - The authors use the word 'believe' in both the first sentence of the abstract and the last sentence of the introduction. In scientific writing, this word should generally be avoided, with a few exceptions. In these cases, 'believe' should be substituted, as the discussion should focus on objective facts rather than personal beliefs.
R: We agree that the word "believe" should be avoided in scientific writing. Consequently, we have replaced it with "evidence indicates" and "probability."
Introduction must present some background of the need of this study and some sentences about the effects of exercise in this period of covid.
R: Thank you for your recommendation. We have adjusted the exclusion criteria to include: Exclusion criteria: older people who did not participate in the assessments carried out in the period in question (2020, 2021, and 2022), individuals with physical limitations to perform the requested evaluations, and use drugs to regulate appetite or psychotropic – Lines: 106-108.
During covid how did you collect data? There were a lot of restrictions how did you acess those people?
R: During both the lockdown period and the gradual resumption of in-person activities, professionals in the field were responsible for developing strategies and implementing measures to protect and ensure the safety of participants and prevent contamination. Prior and individual appointments carried out assessments. Due to the pandemic, participants were advised to follow the World Health Organization (WHO) recommendations. To avoid crowds, appointments were scheduled at specific times, and measures such as the use of masks and constant hand hygiene were implemented. In addition, it was recommended that the exams not be performed if participants presented symptoms of COVID-19, such as fever, sneezing, loss of smell and taste, headache, and body aches. In addition to the individual protection measures for each patient and employee, measures were put in place to ensure distancing between all people and maximum possible hygiene, and suitable places were provided for hand hygiene at all times, use and provision of masks, in addition to keeping the places always ventilated – Lines 128-137.
How can you guarantee that all participants performed the same daily routine (physical activity, nutrition….)
R: Unfortunately, this serves as a limitation of the study. Since we could not assess these parameters with such certainty, we add the following as a limitation of the study: Despite the significant findings, this study has several limitations. One major limitation is the lack of control over participants' daily physical activity. While the study offers valuable insights into the impact of lockdown on physical health, a more comprehensive understanding of the changes would require examining how isolation influenced the type, quantity, and intensity of physical activities performed – Lines 429-434.
Add IC and p values of all variables of Table 2.
R: Ok. We added the Confidence Intervals (CI) and p-values for all the variables in Table 2.
Your discussion needs substancial improvemments. You do not discuss your data with several similar studies of this topic (there are a lot….) This is your weak point of the manuscript submitted. One example "The reduction in VO2peak in 2021 indicates an adverse impact of the pandemic on the aerobic capacity of older people. This decline reflects the body's maximum capacity to use oxygen during exercise and suggests a harmful interference with regular physical activity practice and cardiovascular conditions [26]. The literature indicates that even one year of social isolation in young adults can substantially reduce VO2peak [27]." Where is the discussion of all three- years? Where is the discussion with other studies? You used a study of young adults….
R: Ok. We adjusted this point: The reduction in VO2peak in 2021 indicates an adverse impact of the pandemic on the aerobic capacity of older people. This decline reflects the body's maximum capacity to use oxygen during exercise and suggests a harmful interference with regular physical activity practice and cardiovascular conditions [31]. Silva et al. indicate that some months of lockdown could substantially reduce VO2peak, even with home tele-exercises during COVID-19 [30]. Thus, the reduced VO2peak observed in 2021 refers to low stimulus during the day; these results could be expected since the movement of people has decreased significantly, and even the stimulus cited by da Silva et al. not were enough to promote maintaining this capacity. Telerehabilitation and exercise strategies could be tested to develop new possibilities for maintaining health for the population, especially older people.
Over the past few years, COVID-19 has had a lasting impact on public health, necessitating the development of effective strategies for controlling and treating affected patients [9, 32]. Persistent symptoms such as fatigue, respiratory difficulties, and a reduced ability to carry out daily activities are frequently observed in individuals who have recovered from the acute phase of the infection [10, 11]. Research, such as that conducted by Huang et al. [33], has revealed that a significant number of patients continue to experience debilitating symptoms. During this period, rehabilitation programs focusing on cardiorespiratory and neuromuscular recovery emerged, aiming to help patients return to their daily activities and improve their quality of life [9]. Physical exercise has been increasingly recognized as an effective strategy to mitigate the adverse effects of COVID-19, as highlighted by Jimeno-Almazán et al. [34].
Ongoing monitoring of patients has proven to be essential, as physical fitness is crucial for performing daily activities and maintaining good work conditions. Additionally, the uncertainty surrounding the long-term effects of COVID-19 on various organ systems has underscored the need for constant vigilance and an interdisciplinary approach to patient care [11]. In summary, the evolution of COVID-19 treatment strategies from 2020 to 2022 reflects a progressive shift towards a more integrated and effective management of the disease's long-term effects, focusing not only on restoring patients' physical capacities but also on prevention, as highlighted by Perli et al. [11], ensuring that patients can resume their lives with health and vitality.
Lemos MM, Cavalini GR, Pugliese Henrique CR, Perli VAS, Marchiori GM, Marchiori LLM, et al. Body composition and cardiorespiratory fitness in overweight or obese people post covid-19: a comparative study. Front Physiol. 2022; 21:13:949351. doi:10.3389/fphys.2022.949351
Perli VAS, Sordi AF, Lemos MM et al. Body composition and cardiorespiratory fitness of overweight COVID-19 survivors in different severity degrees: a cohort study. Sci Rep. 2023; 13(1) 17615. doi.org/ 10.1038/s41598-023-44738-8.
Siljá JA, Painelli VDS, Santos IC, Marques DCS, Oliveira FM, Oliveira LP, Branco BHM. No effect of combined tele-exercises and nutritional coaching on anthropometric, body composition or exercise capacity outcomes in overweight and obese women: a randomized clinical trial. Nutr Hosp. 2022; 39(2):329-336. doi:10.20960/nh.03822.
Hedge ET, Hughson RL. Longitudinal assessment of cardiorespiratory fitness and body mass of young healthy adults during COVID-19 pandemic. J Appl Physiol (1985). 2022;133(3):622-628. doi:10.1152/japplphysiol.00253.2022
Chen G., Li X., Gong Z., Xia H., Wang Y., Wang X., et al. (2021). Hypertension as a sequela in patients of SARS-CoV-2 infec-tion. PLOS ONE, 2021; 16(4). doi.org/10.1371/jornal.pone.0250815.
Huang C, Huang L, Wang Y, Li X, Ren L, Gu X, Kang L, Guo L, Liu M, Zhou X, Luo J, Huang Z, Tu S, Zhao Y, Chen L, Xu D, Li Y, Li C, Peng L, Li Y, Xie W, Cui D, Shang L, Fan G, Xu J, Wang G, Wang Y, Zhong J, Wang C, Wang J, Zhang D, Cao B. 6-month consequences of COVID-19 in patients discharged from hospital: a cohort study. Lancet. 2021 Jan 16;397(10270):220-232. doi: 10.1016/S0140-6736(20)32656-8
Jimeno-Almazán A., Buendía-Romero Á Á., Martínez-Cava A., Franco-López F., Sánchez-Alcaraz BJ, Courel-Ibáñez J., et al.Effects of a concurrent training, respiratory muscle exercise, and self-management recommendations on recovery from post-COVID-19 conditions: The RECOVE trial.. J. Appl. Physiol. 2023; 134 ( 1 ), 95–104. doi: 10.1152/japplphysiol.00489.2023
Where is the control group?
R: We insert this point: Another critical limitation is the absence of a control group, which could have significantly strengthened the study by providing a baseline for comparison and enhancing the accuracy of the results. However, considering the daily dynamics during the COVID-19 pandemic, it could be challenging to organize and insert a control group in this study and develop an experimental design with a control group - Lines: 438-443.

Reviewer 2 Report
Comments and Suggestions for Authors
Manuscript Title
" Health-related physical fitness and biochemical parameters in overweight older people during social isolation imposed by the COVID-19 pandemic: a longitudinal and observational study"
1. Summary of the research and overall impression
The present work aimed to study the impact of the social isolation provoked by COVID-19 in Health-related physical fitness and biochemical parameters in overweight older people.
Considering the global impact of social isolation due to COVID-19 on societies, particularly on their health, it is crucial to study in depth which aspects of human health are most affected. This study had that purpose and shows, through the follow-up of a sample of 33 elderly individuals, some of these negative health effects. Since social isolation is not specific to a pandemic, the results of this study may be more relevant. However, the authors failed to make this reference.
The study seems very interesting, but it should have addressed and discussed the lack of control over daily physical activity to understand if there was a change in the type of physical activities, their quantity, and possibly their intensity. Some declines in physical performance may be associated with health changes related to the aging process. Some of the changes are significant but of little magnitude. It is understood that not everything can be evaluated, but the authors should mention these issues.
Despite the well-designed study and the interesting sample, the authors mentioned several other studies in the introduction with similar purposes. I am concerned about the lack of novelty in the study, apart from the Brazilian sample. The authors should emphasize what new insights this study provides.
The manuscript should be revised by a native English-speaking scientist to ensure all the information in the study is clearly understood. For example, in the 3rd sentence of the abstract, “Quantitative, longitudinal, and observational study between 2020, 2021, and 2022,” it seems that “conducted” was omitted.
4rd paragraph 1st sentence : “Concerns after the COVID-19 pandemic due to the emerging numbers of health implications increased sedentary behavior and, consequently, decreased the population's physical fitness”. The sentence is confusing and should be rewritten.
But other examples exist.
2 – Introduction,
The authors use the word 'believe' in both the first sentence of the abstract and the last sentence of the introduction. In scientific writing, this word should generally be avoided, with a few exceptions. In these cases, 'believe' should be substituted, as the discussion should focus on objective facts rather than personal beliefs.
Comments on the Quality of English LanguageThe manuscript should be revised by a native English-speaking scientist to ensure all the information in the study is clearly understood. For example, in the 3rd sentence of the abstract, “Quantitative, longitudinal, and observational study between 2020, 2021, and 2022,” it seems that “conducted” was omitted.
4rd paragraph 1st sentence : “Concerns after the COVID-19 pandemic due to the emerging numbers of health implications increased sedentary behavior and, consequently, decreased the population's physical fitness”. The sentence is confusing and should be rewritten.
But other examples exist.
Author Response
COVER LETTER
Dear Editor,
We would like to thank you for the opportunity to respond to the reviewers. We appreciate their efforts and consideration of all comments, and we believe that their feedback strengthened our manuscript. Our responses to their suggestions are below (answered point by point) and were highlighted in yellow in the original article.
Sincerely,
The authors.
Since social isolation is not specific to a pandemic, the results of this study may be more relevant. However, the authors failed to make this reference.
R: Lockdown during the COVID-19 pandemic not only altered the dynamics of social interactions but also had a profound impact on the mental health of the people who contracted or did not contract the SARS-CoV-2 in different symptoms [4] – lines 45, 46, and 47.
4. Ryal, JJ.; Perli, V.A.S.; Marques, D.C.d.S.; Sordi, A.F.; Marques, M.G.d.S.; Camilo, M.L.; Milani, R.G.; Mota, J.; Valdés-Badilla, P.; Magnani Branco, B.H. Effects of a Multi-Professional Intervention on Mental Health of Middle-Aged Overweight Survivors of COVID-19: A Clinical Trial. Int. J. Environ. Res. Public Health 2023, 20, 4132. doi.org/10.3390/ijerph20054132
The study seems very interesting, but it should have addressed and discussed the lack of control over daily physical activity to understand if there was a change in the type of physical activities, their quantity, and possibly their intensity. Some declines in physical performance may be associated with health changes related to the aging process. Some of the changes are significant but of little magnitude. It is understood that not everything can be evaluated, but the authors should mention these issues.
Thank you for your recommendation. We have added a paragraph with this information in the study's limitations.
Despite the significant findings, this study has several limitations. One major limitation is the lack of control over participants' daily physical activity. While the study offers valuable insights into the impact of lockdown on physical health, a more comprehensive understanding of the changes would require examining how isolation influenced the type, quantity, and intensity of physical activities performed. The absence of post-social isolation monitoring further hinders the assessment of long-term effects and the resumption of physical activities in both indoor and outdoor environments. Additionally, the lack of dietary control and daily physical activity monitoring limits our understanding of changes in eating habits that may have influenced the results - lines 412-421.
Despite the well-designed study and the interesting sample, the authors mentioned several other studies in the introduction with similar purposes. I am concerned about the lack of novelty in the study, apart from the Brazilian sample. The authors should emphasize what new insights this study provides.
R: Thank you for your recommendation. We added some sentences to add this information: This study underscores the critical importance of intervention strategies to enhance muscle strength, cardiovascular health, and metabolic health among older adults, particularly during health crises. The COVID-19 pandemic has illuminated the vulnerability of older individuals to abrupt lifestyle changes, emphasizing the necessity for developing intervention plans that can be swiftly implemented in similar future scenarios – Lines: 407 – 411.
AND
Telerehabilitation and exercise strategies could be tested to develop new possibilities for maintaining health for the population, especially older people – Lines: 360 and 361.
The manuscript should be revised by a native English-speaking scientist to ensure all the information in the study is clearly understood. For example, in the 3rd sentence of the abstract, "Quantitative, longitudinal, and observational study between 2020, 2021, and 2022," it seems that "conducted" was omitted.
Thank you for your feedback. We sent this article again to ensure the accuracy of the English.
4rd paragraph 1st sentence: "Concerns after the COVID-19 pandemic due to the emerging numbers of health implications increased sedentary behavior and, consequently, decreased the population's physical fitness". The sentence is confusing and should be rewritten.
R: To enhance the clarity of the sentence, it has been revised as follows:
"Concerns following the COVID-19 pandemic have grown due to the rising health implications associated with increased sedentary behavior, which has led to a decline in the physical fitness of the population…"
Introduction - The authors use the word 'believe' in both the first sentence of the abstract and the last sentence of the introduction. In scientific writing, this word should generally be avoided, with a few exceptions. In these cases, 'believe' should be substituted, as the discussion should focus on objective facts rather than personal beliefs.
R: We agree that the word "believe" should be avoided in scientific writing. Consequently, we have replaced it with "evidence indicates" and "probability."
Introduction must present some background of the need of this study and some sentences about the effects of exercise in this period of covid.
R: Thank you for your recommendation. We have adjusted the exclusion criteria to include: Exclusion criteria: older people who did not participate in the assessments carried out in the period in question (2020, 2021, and 2022), individuals with physical limitations to perform the requested evaluations, and use drugs to regulate appetite or psychotropic – Lines: 106-108.
During covid how did you collect data? There were a lot of restrictions how did you acess those people?
R: During both the lockdown period and the gradual resumption of in-person activities, professionals in the field were responsible for developing strategies and implementing measures to protect and ensure the safety of participants and prevent contamination. Prior and individual appointments carried out assessments. Due to the pandemic, participants were advised to follow the World Health Organization (WHO) recommendations. To avoid crowds, appointments were scheduled at specific times, and measures such as the use of masks and constant hand hygiene were implemented. In addition, it was recommended that the exams not be performed if participants presented symptoms of COVID-19, such as fever, sneezing, loss of smell and taste, headache, and body aches. In addition to the individual protection measures for each patient and employee, measures were put in place to ensure distancing between all people and maximum possible hygiene, and suitable places were provided for hand hygiene at all times, use and provision of masks, in addition to keeping the places always ventilated – Lines 128-137.
How can you guarantee that all participants performed the same daily routine (physical activity, nutrition….)
R: Unfortunately, this serves as a limitation of the study. Since we could not assess these parameters with such certainty, we add the following as a limitation of the study: Despite the significant findings, this study has several limitations. One major limitation is the lack of control over participants' daily physical activity. While the study offers valuable insights into the impact of lockdown on physical health, a more comprehensive understanding of the changes would require examining how isolation influenced the type, quantity, and intensity of physical activities performed – Lines 429-434.
Add IC and p values of all variables of Table 2.
R: Ok. We added the Confidence Intervals (CI) and p-values for all the variables in Table 2.
Your discussion needs substancial improvemments. You do not discuss your data with several similar studies of this topic (there are a lot….) This is your weak point of the manuscript submitted. One example "The reduction in VO2peak in 2021 indicates an adverse impact of the pandemic on the aerobic capacity of older people. This decline reflects the body's maximum capacity to use oxygen during exercise and suggests a harmful interference with regular physical activity practice and cardiovascular conditions [26]. The literature indicates that even one year of social isolation in young adults can substantially reduce VO2peak [27]." Where is the discussion of all three- years? Where is the discussion with other studies? You used a study of young adults….
R: Ok. We adjusted this point: The reduction in VO2peak in 2021 indicates an adverse impact of the pandemic on the aerobic capacity of older people. This decline reflects the body's maximum capacity to use oxygen during exercise and suggests a harmful interference with regular physical activity practice and cardiovascular conditions [31]. Silva et al. indicate that some months of lockdown could substantially reduce VO2peak, even with home tele-exercises during COVID-19 [30]. Thus, the reduced VO2peak observed in 2021 refers to low stimulus during the day; these results could be expected since the movement of people has decreased significantly, and even the stimulus cited by da Silva et al. not were enough to promote maintaining this capacity. Telerehabilitation and exercise strategies could be tested to develop new possibilities for maintaining health for the population, especially older people.
Over the past few years, COVID-19 has had a lasting impact on public health, necessitating the development of effective strategies for controlling and treating affected patients [9, 32]. Persistent symptoms such as fatigue, respiratory difficulties, and a reduced ability to carry out daily activities are frequently observed in individuals who have recovered from the acute phase of the infection [10, 11]. Research, such as that conducted by Huang et al. [33], has revealed that a significant number of patients continue to experience debilitating symptoms. During this period, rehabilitation programs focusing on cardiorespiratory and neuromuscular recovery emerged, aiming to help patients return to their daily activities and improve their quality of life [9]. Physical exercise has been increasingly recognized as an effective strategy to mitigate the adverse effects of COVID-19, as highlighted by Jimeno-Almazán et al. [34].
Ongoing monitoring of patients has proven to be essential, as physical fitness is crucial for performing daily activities and maintaining good work conditions. Additionally, the uncertainty surrounding the long-term effects of COVID-19 on various organ systems has underscored the need for constant vigilance and an interdisciplinary approach to patient care [11]. In summary, the evolution of COVID-19 treatment strategies from 2020 to 2022 reflects a progressive shift towards a more integrated and effective management of the disease's long-term effects, focusing not only on restoring patients' physical capacities but also on prevention, as highlighted by Perli et al. [11], ensuring that patients can resume their lives with health and vitality.
Lemos MM, Cavalini GR, Pugliese Henrique CR, Perli VAS, Marchiori GM, Marchiori LLM, et al. Body composition and cardiorespiratory fitness in overweight or obese people post covid-19: a comparative study. Front Physiol. 2022; 21:13:949351. doi:10.3389/fphys.2022.949351
Perli VAS, Sordi AF, Lemos MM et al. Body composition and cardiorespiratory fitness of overweight COVID-19 survivors in different severity degrees: a cohort study. Sci Rep. 2023; 13(1) 17615. doi.org/ 10.1038/s41598-023-44738-8.
Siljá JA, Painelli VDS, Santos IC, Marques DCS, Oliveira FM, Oliveira LP, Branco BHM. No effect of combined tele-exercises and nutritional coaching on anthropometric, body composition or exercise capacity outcomes in overweight and obese women: a randomized clinical trial. Nutr Hosp. 2022; 39(2):329-336. doi:10.20960/nh.03822.
Hedge ET, Hughson RL. Longitudinal assessment of cardiorespiratory fitness and body mass of young healthy adults during COVID-19 pandemic. J Appl Physiol (1985). 2022;133(3):622-628. doi:10.1152/japplphysiol.00253.2022
Chen G., Li X., Gong Z., Xia H., Wang Y., Wang X., et al. (2021). Hypertension as a sequela in patients of SARS-CoV-2 infec-tion. PLOS ONE, 2021; 16(4). doi.org/10.1371/jornal.pone.0250815.
Huang C, Huang L, Wang Y, Li X, Ren L, Gu X, Kang L, Guo L, Liu M, Zhou X, Luo J, Huang Z, Tu S, Zhao Y, Chen L, Xu D, Li Y, Li C, Peng L, Li Y, Xie W, Cui D, Shang L, Fan G, Xu J, Wang G, Wang Y, Zhong J, Wang C, Wang J, Zhang D, Cao B. 6-month consequences of COVID-19 in patients discharged from hospital: a cohort study. Lancet. 2021 Jan 16;397(10270):220-232. doi: 10.1016/S0140-6736(20)32656-8
Jimeno-Almazán A., Buendía-Romero Á Á., Martínez-Cava A., Franco-López F., Sánchez-Alcaraz BJ, Courel-Ibáñez J., et al.Effects of a concurrent training, respiratory muscle exercise, and self-management recommendations on recovery from post-COVID-19 conditions: The RECOVE trial.. J. Appl. Physiol. 2023; 134 ( 1 ), 95–104. doi: 10.1152/japplphysiol.00489.2023Where is the control group?
R: We insert this point: Another critical limitation is the absence of a control group, which could have significantly strengthened the study by providing a baseline for comparison and enhancing the accuracy of the results. However, considering the daily dynamics during the COVID-19 pandemic, it could be challenging to organize and insert a control group in this study and develop an experimental design with a control group - Lines: 438-443.
